# Treetop Detection in Mountainous Forests Using UAV Terrain Awareness Function

Orou Berme Herve Gonroudobou [1], Leonardo Huisacayna Silvestre [1], Yago Diez [2], Ha Trang Nguyen [1] and Maximo Larry Lopez Caceres [1,*]

1   Faculty of Agriculture, Yamagata University, Tsuruoka 997-8555, Japan;
    gonroudobou0014@tds1.tr.yamagata-u.ac.jp (O.B.H.G.); leo.hs334@gmail.com (L.H.S.);
    nguyen0005@tds1.tr.yamagata-u.ac.jp (H.T.N.)
2   Faculty of Science, Yamagata University, Yamagata 990-8560, Japan; yago@sci.kj.yamagata-u.ac.jp
*   Correspondence: larry@tds1.tr.yamagata-u.ac.jp

**Abstract:** Unmanned aerial vehicles (UAVs) are becoming essential tools for surveying and monitoring forest ecosystems. However, most forests are found on steep slopes, where capturing individual tree characteristics might be compromised by the difference in ground sampling distance (GSD) between slopes. Thus, we tested the performance of treetop detection using two algorithms on canopy height models (CHMs) obtained with a commercial UAV (Mavic 2 Pro) using the terrain awareness function (TAF). The area surveyed was on a steep slope covered predominantly by fir (*Abies mariesii*) trees, where the UAV was flown following (TAF) and not following the terrain (NTAF). Results showed that when the TAF was used, fir trees were clearly delimited, with lower branches clearly visible in the orthomosaic, regardless of the slope position. As a result, the dense point clouds (DPCs) were denser and more homogenously distributed along the slope when using TAF than when using NTAF. Two algorithms were applied for treetop detection: (connected components), and (morphological operators). (connected components) showed a 5% improvement in treetop detection accuracy when using TAF (86.55%), in comparison to NTAF (81.55%), at the minimum matching error of 1 m. In contrast, when using (morphological operators), treetop detection accuracy reached 76.23% when using TAF and 62.06% when using NTAF. Thus, for treetop detection alone, NTAF can be sufficient when using sophisticated algorithms. However, NTAF showed a higher number of repeated points, leading to an overestimation of detected treetop.

**Keywords:** canopy height model; terrain awareness; unmanned aerial vehicle; treetop detection

## 1. Introduction

Forests on steep terrain are found all over the world, and in fact are more abundant than those in flat areas, where they have already been cut for human settlements or agricultural activities. One of these cases is Japan, where most of the 68% forest coverage is found on steep mountains slopes, ranging from 35 to 45 degrees [1]. Forest surveys under these conditions limit the access to a small set of sample plots, where ground measurements can be conducted, constraining the understanding of the larger remainder. Until recent years, satellite images were widely used tools to capture large forest areas with a reasonable level of detail, depending on the resolution of the images and, most importantly, their cost. In these images, the uneven terrain of mountains is taken for granted, and the problems of resolution and errors caused by the slope are accepted. Over the years, studies have dealt with the design of terrain-following applications for aircrafts, and it is only recently that this function has become an important issue for flight plans of UAVs (unmanned aerial vehicles) over steep terrain. Most studies using UAVs considering terrain-following flights have focused on the detection of geomorphic changes such as volcanos, landslides, glaciers, or gorges in different parts of the world [2–6], but to our knowledge none has focused on forest characteristics in steep terrain. Some of the authors of these studies have described a

clear strategy to deal with the issue of following the terrain, e.g., [6] tackled the problem of slopes by flying automatically and manually in stripes at different altitudes.

The result of flying at different altitudes in separate flight plans, or of flying manually to keep the same distance from the UAV to the ground along slopes, is that in the first case blank spots are produced in the orthomosaic and, in the second case, keeping the overlapping of successive images is compromised. This is especially relevant when the focus is the detailed information of dendrometric parameters of forests, since tree characteristics appear different depending on their position on the slope. Recent UAV applications in forestry research have shown the immense potential of the very-high-resolution images for capturing individual tree details [7–9]; however, the issue of following the terrain has not been addressed, even though the differences might have a significant effect on the perception of the tree canopy area, the density of the point cloud, and the canopy height model at different positions along the slope. No information concerning following the terrain can be found in either the thorough review of UAVs in forestry [10] or the review of applications of deep learning in forestry using UAV imagery [11]. The extraction of more detailed information is important for the precise estimation of forest dendrometric parameters [12], forest health [13], gaps, and forest species composition [14].

Thus, UAV flight plans that include the terrain awareness function (TAF) will maintain the same ground sampling distance (GSD) and capture the characteristics of not only the treetops of dominant trees, but also those in the co-dominant and even suppressed layers within the forest structure. Ref. [15] showed that the precision using the TAF is enhanced when users create their own digital surface model (DSM) of a given slope to guide the UAV, as these data are input in the flight plan of the UAV when following the terrain. However, this cannot be done in most of low-cost commercial UAVs, such as the Mavic 2 Pro, for which applications such as DroneDeploy provide the following awareness function using the available global digital terrain data (SRTM, etc.).

Individual tree-level information is crucial for forest management, with tree height being one of the most important parameters for dendrometric calculation. UAV-processed data usually need to be annotated to point out the location or area of interest. One typical annotation method is treetop, pointing at the highest elevation value. Since manual annotation can be time-consuming—especially for large areas—numerous studies have attempted to automate individual treetop detection. Tree crown shape and terrain complexity affect treetop detection [16], since systematic distortion caused by slope terrain normalization reduces the performance of the treetop detection algorithm [17]. Flight at a constant height over a slope increases treetop displacement during normalization. Several approaches—including artificial intelligence methods such as convolutional neural networks [18]—have been used for treetop detection, but none has focused on treetop detection when following the slope.

The application of computer vision techniques—such as the local maxima algorithm on the canopy height model (CHM) [9,19–21] or on the dense point cloud (DPC) to automatically detect treetops within a forest stand—has been used in studies in mountainous terrain. However, in these cases, the image collection did not follow the terrain, and it is possible that the quality of the DPC and the CHM could have not reached their maximum potential, as their heterogeneity within the surveyed area was not taken into consideration. Thus, the treetop detection algorithm might have a good performance in some areas of the CHM but a poor performance in others [21]. This issue was clearly observed in [9], where co-dominant trees observed in the orthomosaic were not found in the CHM, mainly because of the lack of data in the DPC for lower trees. Thus, we hypothesized that DPCs generated from UAV-acquired images using the TAF would improve the performance of the treetop detection algorithm which, in turn, would have a positive impact on forest management in mountainous areas. Therefore, the aims of this study were (1) to compare the difference in the quality of DPCs and CHMs produced by a Mavic 2 Pro on a slope covered by fir trees using the TAF, and (2) to evaluate the performance of two treetop detection algorithms on the CHM when the terrain is followed (TAF) and when it is not (NTAF).

## 2. Materials and Methods

### 2.1. Study Site

We conducted this study in the Zao Mountains (Figure 1) a composite stratovolcano cluster in southeastern Yamagata Prefecture (140°24′39.224″ E, 38°9′0.327″ N) on a 20° slope covering an area of 3.8 ha. The fir stand in the slope has a density of 117 trees/ha, and is dominated by mature Maries' fir (*Abies mariesii*) trees mixed with deciduous broadleaf species (e.g., *Acer* spp., *Fagus crenata*, *Quercus mongolica*, *Sorbaria sorbifolia*). Fir is a highly valuable tree in Japan because of its obvious ecological function, but mainly as a tourist attraction, as they form the famous "Snow Monsters" in winter when they are covered completely by snow. In recent years, bark beetle attacks have seriously affected fir trees' health [22].

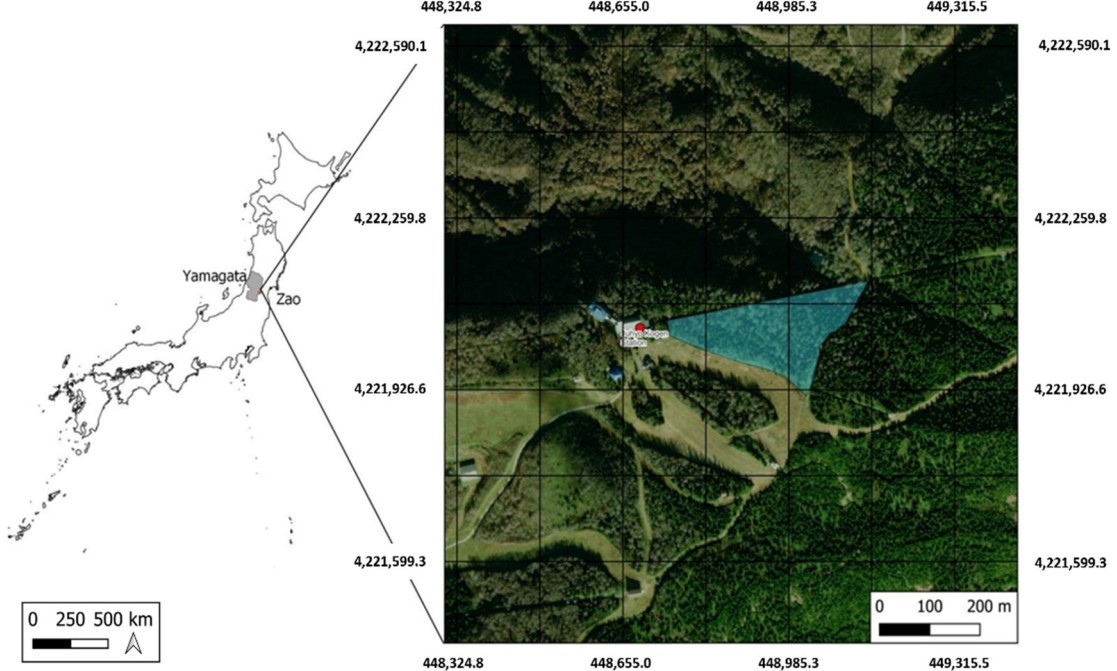

**Figure 1.** Location of the study site in the Zao Mountains next to *Juhyo Kogen* ropeway station. The fir trees are mixed with deciduous broadleaf species on the selected slope.

### 2.2. Unmanned Aerial Vehicles and Flight Plans

We collected the data using a Mavic 2 Pro DJI quadcopter drone equipped with a height-definition RGB digital camera of 1 inch CMOS 20 MP effective pixels. The L1D-20C camera of the Mavic 2 Pro is equipped with a 77 degree viewing angle lens, numerical shutter, and Hasselblad Natural Color Solution (HNCS) that can reproduce good detail color (10-bit) for a 5472 × 3648 image size. The pictures are georeferenced with the drone's onboard positioning global navigation satellite system (GPS and GLONASS).

A digital elevation model (DEM) is necessary to enable the UAVs to follow the terrain on slopes. For automated flight missions, we used the "DroneDeploy" application which, in contrast to the original DJI application "DJI GS PRO", offers a terrain awareness function using an online Mapbox-optimized dataset based on NASA's SRTM elevation grid.

Sets of RGB images were collected on October 5, in autumn, to better distinguish the spectral contrast of fir trees from the surrounding senescing colors of the deciduous trees. We flew two missions: one with TAF and one with NTAF, using the same parameters. The settings were as follows: 90 m flying altitude, and 85% front and side overlapping to create structure-from-motion (SfM) 3D models. This setting led to a ground resolution of 1.98 and 2.75 cm, and an average distance from the cameras to the sparse cloud points of 95.8 and 127 m along the slope, for TAF and NTAF, respectively. The height above ground

varied from 79.5 to 100.5 m for TAF, and from 85.0 to 157.0 m for NTAF. We collected 281 to 285 images with a resolution of 5472 × 3648 pixels for each flight.

### 2.3. Image Analysis

DPCs, DSMs, and orthomosaics were generated for post-processing analysis tasks using Metashape Professional v1.7.4 (Agisoft LLC, Saint Petersburg, Russia). The DPC—a desegregation of images contained in a set of millions of points with high values of spatial resolution (x, y, z)—is the basis of any form of digital image's processed data. The site's point density was assessed using DPCs from both flights focusing on the same region of interest (clip by the same georeferenced polygon) after removing duplicated cloud points. DPCs were normalized (Figure 2) using an execute command batch file run in the LIDAR data analysis software FUSION/LDV v4.10 (McGaughey, R.J., US Department of Agriculture, Forest Service, Pacific Northwest Research Station: Seattle, WA, USA) [23], following three standard steps (i.e., ground filter, grid–surface create, and clip data) and smoothing parameters. We used the software Global Mapper v21.1.0 (Blue Marble Geographics, Hallowell, Maine, United States)—a cutting-edge GIS software package—to generate elevation grids based on nDPCs (normalized DPCs).

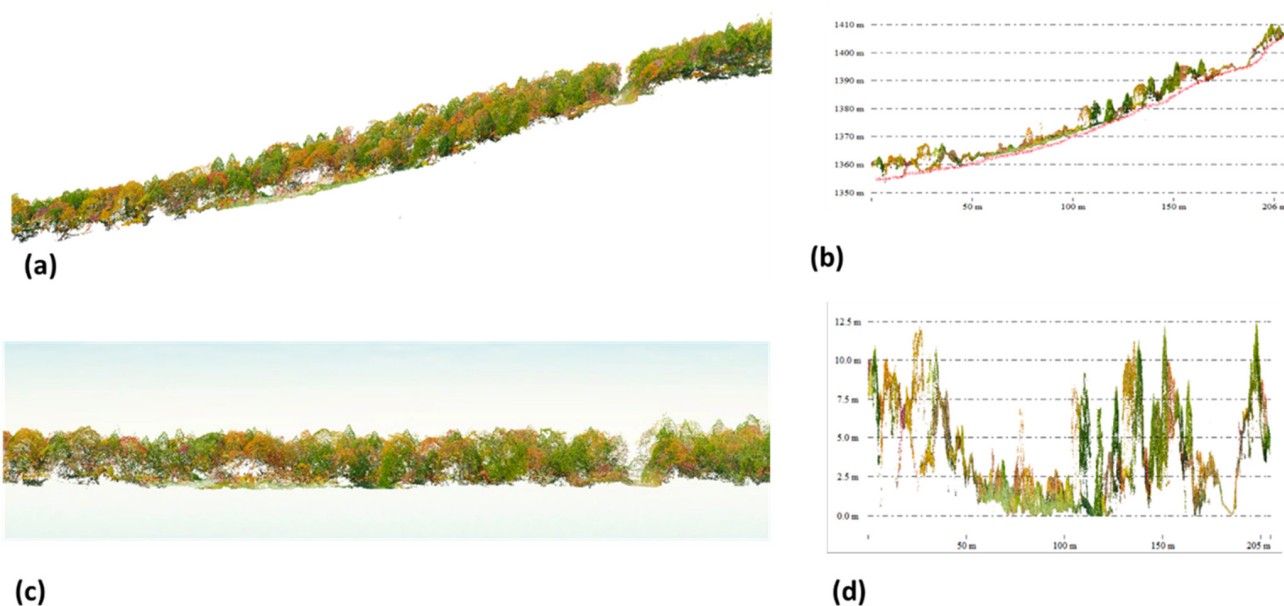

**Figure 2.** DPC normalization: (**a**) DPC; (**b**) filtered ground points based on a digital terrain model; (**c**) normalized points; (**d**) height above ground.

The form of data that was used for more direct treetop detection in our study was CHM—a grayscale 2D above-ground elevation model obtained from the traditional difference (made using FUSION/LDV software) between the digital surface model (DSM) and the digital terrain model (DTM).

The annotations (fir treetops) on the RGB orthomosaic were made using QGIS v3.22. The vectors' shape files were rasterized, and the output images were exported in PNG format. The generated CHM files were stored in TIFF format and used as data inputs for the treetop detection algorithms. In the final step (data validation), the results of the treetop detection algorithms were compared to the manual annotations (Figure 3).

### 2.4. Problem Definition

The GSD varies along the slope, leading to heterogeneous proportion and resolution of objects. We divided our region of interest into three areas (bottom, middle, and upper), following a 10 m terrain contour line interval (Figure 4), in order to assess the effect of fluctuations in the UAV's above-ground flying height on the data quality. The average

GSD, when the TAF was used, for the bottom, middle, and upper areas of the slope was 2.06, 1.96, and 1.99 cm/px, respectively, with a range of 0.1 cm. In contrast, when the NTAF was used, the GSD in the same regions was 3.35, 2.91, and 2.49 cm/px, respectively, with a range of 0.86 cm.

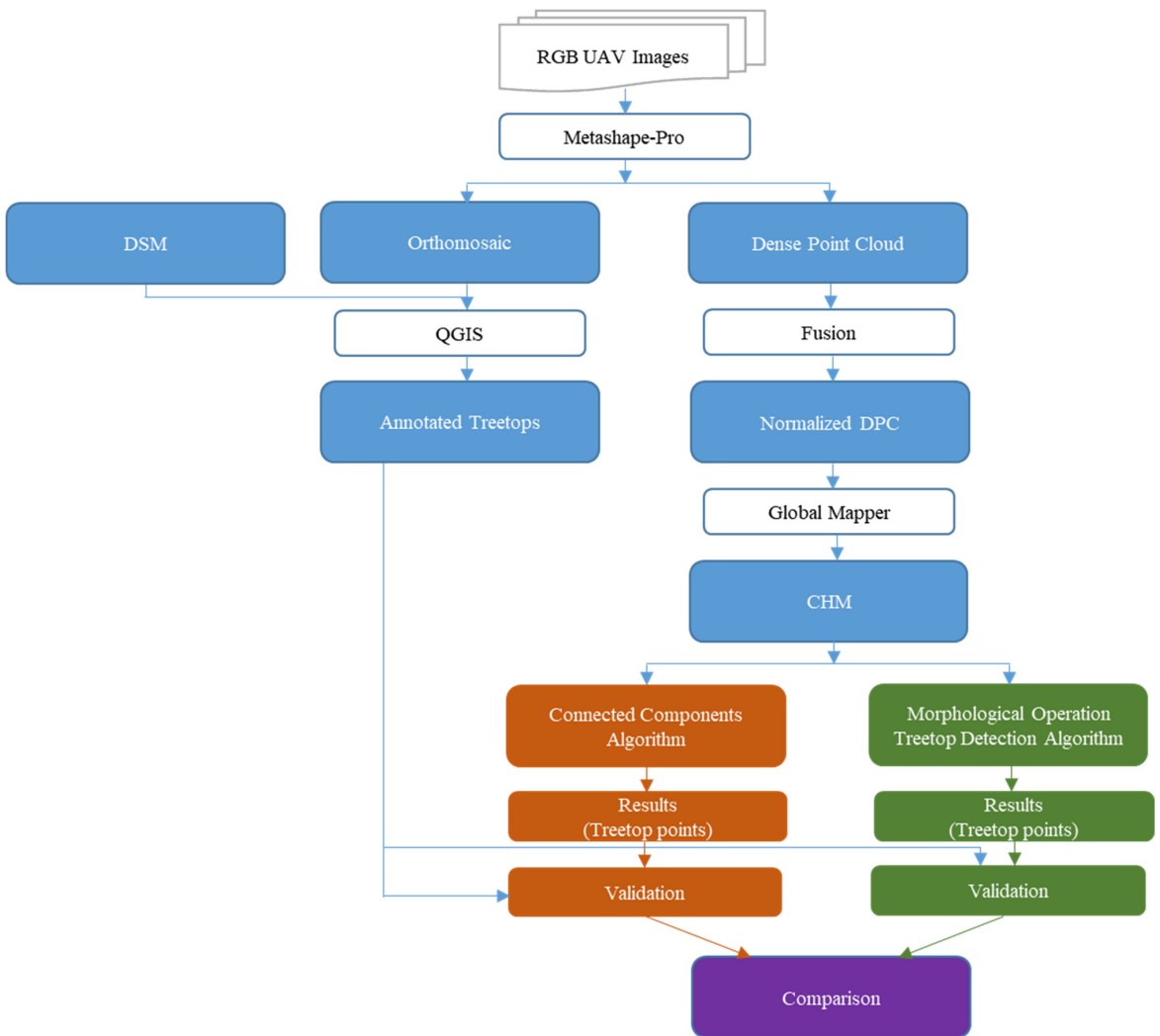

**Figure 3.** Study workflow: processed data (blue), treetop detection algorithms' process (orange and green), algorithm results comparison (purple), and commercial software used (white box).

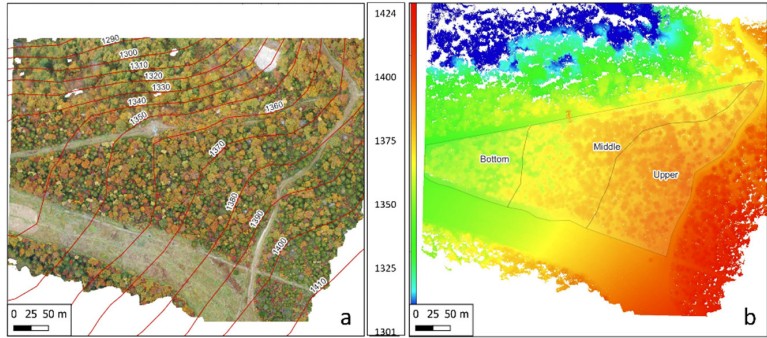

**Figure 4.** (**a**) Ten-meter contour line on the RGB orthomosaic of the study site, and (**b**) the study site divided into three areas related to their altitude: bottom (1330 m–1350 m), middle (1350 m–1370 m), and upper (1370 m–1390 m).

### 2.5. Treetop Detection Algorithms

#### 2.5.1. Connected Components

In order to take full advantage of the precise data measurements represented in the CHM, a GeoTIFF data format with floating components was used, allowing us to encode altitude values in millimeters. In order to detect treetops in the CHM, we used a modification of the algorithm described in [9] to adapt it to the higher quality of the CHMs used in the present work, and for detecting only fir trees. Treetops can be formalized as local maxima in the 2.5D canopy surface discretized in the CHM files. Consequently, in this algorithm, we performed a series of local searches (Figure 5), allowing us to minimize memory requirements by using a "sliding window approach".

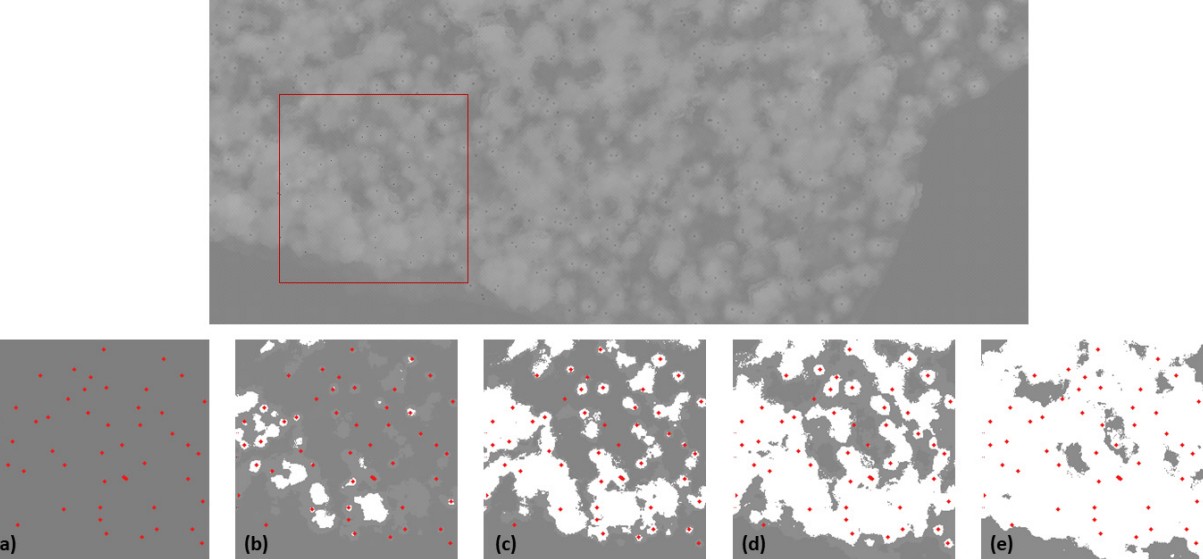

**Figure 5.** Fixed-side-length windows over the CHM sliding on different heights from above with (**a**) expert-annotated treetop positions, then (**b**) the initial tallest region appearing, (**c**,**d**) new appearances and, finally, (**e**) the region's highest intensity marked as treetops.

#### Sliding Window Approach

A bounding box for each CHM file was considered, and a partition of this bounding box into fixed-side-length "s" "windows" was used. This process is analogous to picturing a single "s" side-length window that slides over the bounding box of the dataset being processed (Figure 5). At each window position, we determined the local maxima corresponding to treetops, characterized as follows:

- Treetops are the highest points in their neighborhood.
- Treetops are surrounded by lower points forming the rest of the tree canopy, such that, when looked at from above, they would be separated from other treetops (at least in their upper part). Fir tree canopies are roughly conical, so a set of fir trees seen from above can be pictured as a set of overlapping circles, with the treetop at the center of the circles. As trees have different heights, it is difficult to automatically assess where each tree starts and where it ends.

#### Treetop Determination

For each position of the sliding window, we carefully considered the altitudes of the pixels in decreasing order, and kept track of newly appearing trees. First, only a very narrow band of the pixels corresponding to the tallest trees in the window was considered. The pixels not in this band were set to 0. Consequently, in this initial band, only the topmost part of the tallest trees would appear as disjoint regions. We computed these regions (hereafter "connected components") using a DAG labeling algorithm [24] implemented in the "ConnectedComponentsWithStats" method of the OpenCV library. We ignored the

connected components whose area was under a certain area threshold (minPoints) to avoid being misled by noise or possible image artifacts. Several values were considered for this parameter, to finally set a value of one-fifth of the acceptable location error between treetops ($\varepsilon/5$). We found the highest point in each of the components that was large enough, and labeled it as a treetop. Whenever a new "large enough" connected component appeared, the highest value of that component in the CHM would be designated as a new treetop, and a point would be assigned to it.

This process continued within the window until all pixel intensities were considered. At each new step, the band of intensities considered was widened, and the connected components in the resulting thresholded part of the CHM window were considered. For each connected component at each step, we first determined whether or not it was a newly appearing component by checking whether any of the already-detected treetops belonged to it. For newly appearing connected components, a new treetop would be detected at their highest point. Figure 5 presents a visual example of how this process develops. In the bottom-left part of Figure 5a, only the treetops are depicted as red points. In Figure 5b–e, we show how every time we widen the band of pixels being considered, more treetops can be detected.

Once all of the intensity values at one particular window location had been considered, the window was shifted to a new location. In order to avoid missing treetops "between windows", they had a small (5%) overlap between them.

All code was implemented in the Python programming (pseudo code in Algorithm 1) language [25], using the OpenCV library [26], and is available from the authors of the paper on demand.

---

**Algorithm 1.** Treetop determination and sliding window description as pseudocode

---

**1.1** Find_Tops_Connected_Components(CHM,minPoints,step)

---

$W \leftarrow AAWindows(CHM)$     ▷ Set of axis-aligned "windows" sliding over the CHM
$Tops \leftarrow \varnothing$     ▷ "Tops" initialized as empty list
**for** $w \in W$ **do**
$currentTop \leftarrow Process\_Window(w, minPoints, step)$     ▷ Find tops in this window
**end for**
**if** currentTops $\neq \varnothing$ **then**
extend_list(Tops,currentTops)     ▷ Add tops to those of previous windows
**end if**
**return** tops

---

**1.2** Process_Window(w,minPoints,step)

---

$wTops \leftarrow \varnothing$     ▷ Tops in this window initialized as empty list
$maxAlt \leftarrow Maximum(w)$
$expTH \leftarrow maxAlt - step$     ▷ Explore altitude values from expTH upwards
**while** expTH > 0 **do**
$thWindow \leftarrow Threshold(w,expTH,maxAlt)$     ▷ Delete altitudes < expTH
$C \leftarrow ConnectedComponentsWithStats(thWindow$     ▷ Detect treetop candidates
**for** c $\in$ **C do**
if Area(c) > minPoints **then**     ▷ Component big enough to contain top
if NoTopInComponent(c,wTops) **then** ▷ No previous top is in this component
$Top \leftarrow location(max(c))$
wTops.Append(top)     ▷ Added new top to list of tops in the window
**end if**
**end if**
**end for**
$thWindow \leftarrow thWindow - step$     ▷ Update loop condition
**end while**
**return** wTops

---

Connected Components Modification

In [9], the algorithm made two passes over the CHM in order to account for smaller trees that could be detected in the lower heights of the CHM. As we were only looking to detect fir trees, and not to include deciduous trees, we modified the algorithm to perform a single pass. Moreover, the shapes of fir trees are more clearly defined than those of deciduous trees, so the minimum number of points needed to consider a detected connected component as a treetop was lowered, and the rate at which new height values were added for consideration was increased.

### 2.5.2. Morphological Operations

This algorithm used computer vision techniques to erase the borders (i.e., areas close to the floor) of local regions of the CHM. We used the fact that treetops are usually located in the middle of circular regions at high local altitude. By repeatedly erasing the borders of the local regions in the CHM, we could isolate most of the trees and find their treetops (Figure 6). Morphological operators were used to isolate pixels at the maximum height of their local area, and were applied to the whole CHM.

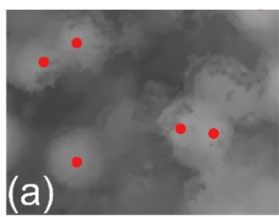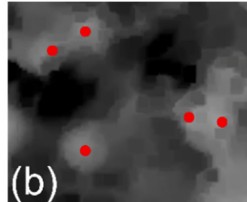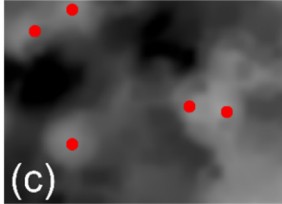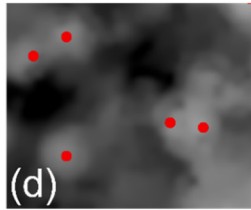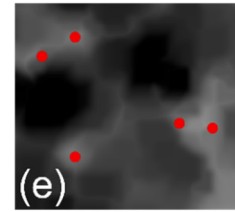

**Figure 6.** Image morphological operations cycle on (**a**) CHM with manual treetop annotation, (**b**) erosion, (**c**) Gaussian blur, (**d**) dilation, and (**e**) erosion.

The steps of the algorithm were as follows:

- First, the lower-altitude pixels were filtered out using global thresholding of the image.
- The resulting grayscale image was eroded using an elliptical kernel to remove the borders of groups of pixels, with the goal of separating trees.
- Then, a Gaussian blur filter was used to smooth individual tree canopies.
- Dilation was then used in the blurred image to make the highest pixels occupy a wider area.
- To ensure that no previously separated regions had been reunited, bitwise comparison was used to compare the dilated and non-dilated images.
- After that, cycles of erosion + bitwise AND were used to isolate smaller and smaller regions. This step had to be carried out with particular care, as too many erosion operations may totally wipe out small trees.
- The groups of pixels left in the image were identified as treetop regions, and the central point in each one was considered as a detected treetop.

Algorithm 2 presents a pseudocode version of the algorithm. All code was implemented in the Python programming language (PYTHON) using the OpenCV library (OPENCV), and is available from the authors of the paper on demand. The different parts of this algorithm were implemented using OpenCV's morphological operations mode, which performs local bitwise operations by applying a small morphological kernel (in this case, an elliptical one) at all possible positions of the CHM image.

### 2.6. Treetop Detection Validation

A series of validation metrics were calculated in order to assess the accuracy of the treetop detection algorithms. The result of each algorithm was a set of 2D points, where three metrics were applied to assess the effectiveness of automated treetop detection based on the expert annotation (ground-truth point). The number of trees annotated on the orthomosaic was 464.

---

**Algorithm 2.** Morphological operation and treetop detection pseudocode.

---

Find_Tops_Morphological_Operators(CHM,minAlt,numIters)

---

*thCHM←Threshold(CHM, minAlt, Maximum(CHM)*
▷ Pixels under minAlt become black
*ero ← Erosion(thCHM, eKernel)*          ▷ Elliptical kernel erosion (isolate regions)
*blurr ← GaussianBlurr(ero, eKernel)*        ▷ Gaussian blur (smooth canopies)
*dil ← Dilation(blurr, eKernel)*
*comp ← (blurr > dil)*
▷ Keep only pixels that have increased in value because of dilation
*It ← 0*
*Im ← comp*
**while** *it < numIters* **do**
*newEro←Erosion(im, eKernel)*
*im = newEro & im*
▷ Further isolate regions erosion + logical AND
*it ← it + 1*                  ▷ Update loop iterator
**end while**

---

**Matched ground-truth points percentage (m%):** The aim of this criterion was to check how many treetops were correctly detected. The mean fir tree crown radius from the study site orthomosaic was 2 m, but we used different margins of error (e.g., 1, 1.5, 2 m) for thorough validation. For the rest of the paper, we also refer to this margin of error as "$\varepsilon$". The points within the considered radius value threshold of a ground-truth point were considered to be "matched".

**Repeated ground-truth points percentage:** In this step, we also computed the percentage of ground-truth points that were matched more than once. This criterion indicated the source of the prediction overestimation more thoroughly. A high number indicated a difficulty to separate individual treetops, and a low number indicated erroneous points being detected in the outer parts of the tree canopies.

**Counting measure (cnt):** This represents the difference between the number of trees present in the CHM "n" and the number of treetops detected "k" cnt = (n − k). Consequently, negative values indicate that the algorithm overestimated the number of trees, while positive values indicate underestimation.

Even though it would be possible to define a matched predicted point as a true positive prediction and an unmatched one as a false positive, and to use these labels to use well-established metrics such as sensitivity, specificity, and F-score [27], this definition would not take into account multiple matchings from predicted points to ground-truth points, or vice versa. Taking into account that tree counting is an important problem in our application scenarios, we decided to use the aforementioned measures (three criteria) to target broader possibilities of treetop counting.

## 3. Results

### 3.1. TAF vs. NTAF Datasets—Qualitative Evaluation

The orthomosaics made with images collected using the TAF and NTAF showed different results concerning the detailed fir tree characteristics—mainly a deep view of tree canopies along the slope. When the TAF was used, the tree canopy area (lower branches) was clearer in the bottom and upper areas of the slope (Figure 7a). Conversely, when the NTAF was used, the details of the lower tree layers within the fir stand were missed at the bottom area of the slope, because of the higher GSD (Figure 7b). In general, tree canopies' shape did not show any distortion after stitching images collected at different altitudes in order to assemble the orthomosaic when the TAF was used. This is relevant because the treetop annotation might not be precise in the actual center of the tree, where the canopy shape has been affected by the steepness of the slope.

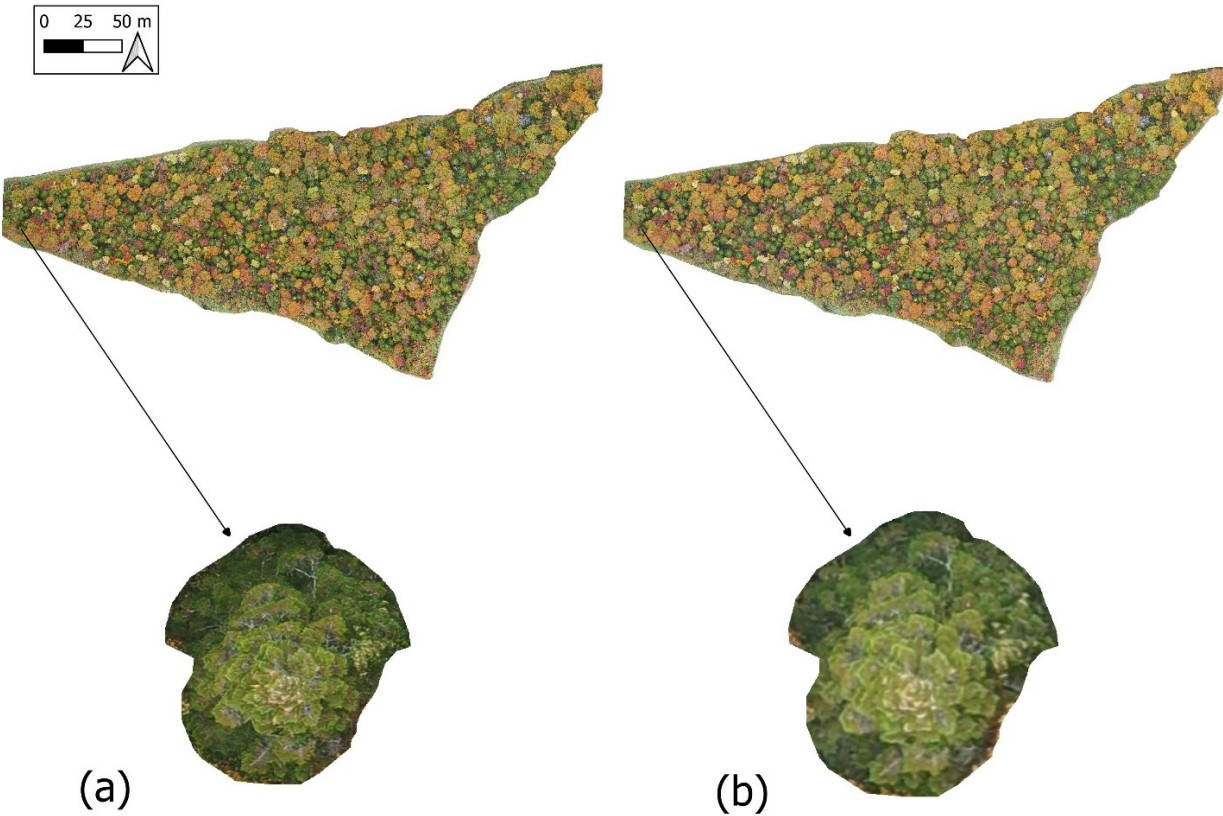

**Figure 7.** Regions of interest and fir tree canopy difference in the bottom area of the orthomosaic: (**a**) when using the TAF, the lower branches and the shape are clearer, and (**b**) when not, fewer details are visible.

### 3.2. Dense Point Clouds (DPCs) and Canopy Height Model (CHM)

The density of the DPC significantly increased and its points were distributed uniformly along the slope when the TAF was used (Figure 8). The total number of points generated in the study site was 15,099,519 when using TAF and 8,599,946 when flying at a constant height (NTAF). The number of filtered ground points at the bottom and middle areas was 381,048 and 33,024, respectively, while when the TAF was used the number of points was 747,184 and 57,432, respectively (Figure 8. In general, trees' dimensions along the slope when the TAF was used were more homogeneous than when it was not used. There were no blank spots in the orthomosaic obtained with the TAF flight, despite using the commercially available SRTM.

### 3.3. Treetop Detection

Considering a margin of error of 2 m (two meters between a predicted point and an annotated treetop) as an acceptable matching error, Algorithm 1 obtained results slightly close to 90% for both datasets, reaching a maximum of 90.81 matching percentage using the TAF and 89.91% for the NTAF, while Algorithm 2 detected about 81% of the existing points using the TAF dataset and about 70% using the NTAF dataset (Table 1). Algorithm 1 and Algorithm 2 show an increasing repeated percentage as the margin of error increases, and less repetition when using the TAF. Algorithm 1 repeated 1.04% of matched points at $\varepsilon = 1$ m when using the TAF, while Algorithm 2 showed a maximum repetition of 7.17% when using the NTAF. This illustrates how by using the TAF we can obtain predicted points that are closer to the ground-truth points.

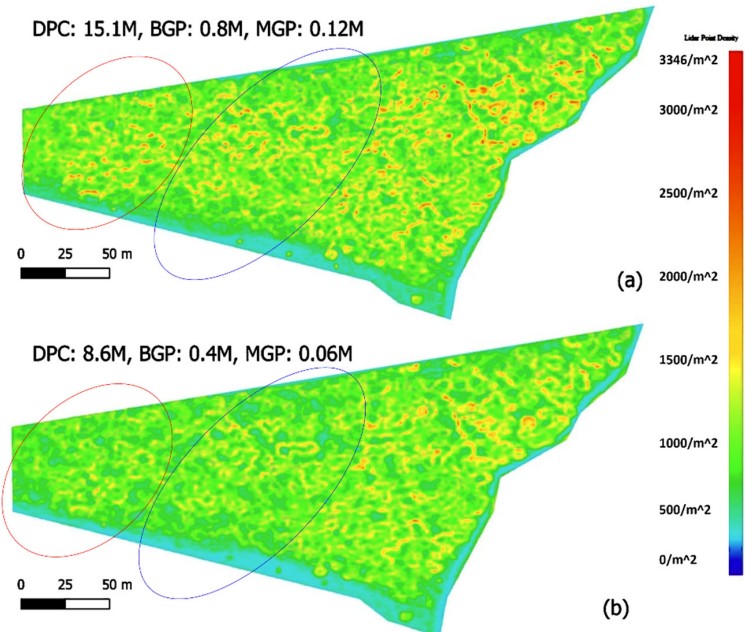

**Figure 8.** Distribution of the DPC when applying the (**a**) TAF and (**b**) NTAF. The difference in DPC ground point numbers in millions (M), bottom-area (red circle) ground points (BGPs), and middle-area (blue circle) ground points (MGPs) was significant when using the (**a**) TAF and (**b**) NTAF.

**Table 1.** Treetop detection validation: Results of the matching percentage and repeated percentage. This table shows the results of the two algorithms when using the TAF and NTAF, following three crown-radius parameters.

| Margin of Error | | 1 m | 1.5 m | 2 m |
|---|---|---|---|---|
| Matching% | | | | |
| Algorithm 1 (connected components) | TAF | 86.55 | 88.79 | 90.81 |
| | NTAF | 81.8 | 87.5 | 89.91 |
| Algorithm 2 (morphological operation) | TAF | 76.23 | 79.37 | 81.17 |
| | NTAF | 62.06 | 68.2 | 70.39 |
| Repeated% | | | | |
| Algorithm 1 (connected components) | TAF | 1.04 | 2.02 | 3.21 |
| | NTAF | 1.07 | 4.51 | 6.59 |
| Algorithm 2 (morphological operation) | TAF | 1.18 | 3.95 | 6.08 |
| | NTAF | 1.77 | 4.82 | 7.17 |

The counting measure (cnt) showed an overestimation of the number of detected treetops of up to 10% for Algorithm 1, while Algorithm 2 tended to underestimate the number of treetops (Table 2).

**Table 2.** Difference between ground-truth treetops and detected treetops (cnt), where positive values indicate detected treetop underestimation and negative values indicate overestimation.

| | | Count Measure (cnt) |
|---|---|---|
| Algorithm 1 (connected components) | TAF | −10.99 |
| | NTAF | −1.75 |
| Algorithm 2 (morphological operation) | TAF | 19.66 |
| | NTAF | 33.11 |

## 4. Discussion

UAV image collection presented a remarkable flexibility in adapting to the environmental and topographical characteristics of the forest site that was the focus of our investigation. Most of the UAV missions in the reviewed literature have focused on surveying the terrain, rather than on the vegetation cover, and to the best of our knowledge none has focused on the effect of the terrain on the forest characteristics obtained from orthomosaics. When the terrain is the main objective of a UAV mission, the optical axis of the camera is typically set oblique to the ground [15]. However, since our focus was on the treetops distributed on a slope, we set the optical axis of the camera vertical ($-90$) to the ground since, regardless of the slope angle, trees along the slopes had a vertical direction towards the camera. The results of using the Mavic 2 Pro—one of the most affordable and versatile UAVs on the market—together with DroneDeploy software, allowed flights that followed the terrain with high precision, unlike the results found in [28], where it was found that TAF flights were prone to creating blank spots in the orthomosaic. Thus, the M2P flew smoothly along the slope, keeping a stable height and GSD, and spikes of altitude changes during flight were not observed, in contrast to the findings of [15].

From the point of view of forestry, the increase in the number of points in the DPC with TAF flights provides a more accurate depiction of individual tree structures. The sharp increase in the number of points captured in the tree canopy should be taken with care, because there is more duplicate information generated when processing the data. Nevertheless, the higher density of the DPC is important not only for treetop detection, but also for accurate detection of tree canopy characteristics that can be used for the precise evaluation of, for example, forest health [7,9], forest fire disturbance [29], or the estimation of dendrometric parameters [30]. This is especially relevant for fir forests in the Zao Mountains in Japan because, as shown in [22], the rate of single fir tree defoliation can be used as a proxy, and the results of using the TAF will contribute to a higher precision of tree canopy evaluation of the forest stand along the slope.

A higher feature number (denser DPC) decreases the 3D point triangle's face network interpolation effect and, therefore, results in a better-balanced elevation grid (CHM). Additionally, more filtered ground points enhance the CHM calculation accuracy. Consequently, in this study, we were able to use CHMs (in the TAF case) that were much denser locally than in previous studies using UAVs to survey forest ecosystems, where only the NTAF was used.

Two algorithms were used to perform automatic treetop detection on the CHMs using the TAF and NTAF. Both algorithms performed better when using the TAF, as shown in Table 1. When predicted treetops were allowed to deviate no further than one meter ($\varepsilon = 1$) from the ground-truth treetops, Algorithm 1 detected 86.55% and 81.80% (TAF and NTAF, respectively) of the treetops, while Algorithm 2 detected 76.23% and 62.06% (TAF and NTAF, respectively). These numbers also indicate that the use of the TAF facilitates the prediction of a larger number of treetops that are closer to the ground truth.

The percentage of matched points grew sharply for both algorithms when $\varepsilon$ was increased, especially for the TAF dataset. This was especially clear for Algorithm 2, which was less tailored to the current data. With Algorithm 1, we were able to reproduce the results of [11]—specifically, 89.6% for healthy fir trees and 90.7% for sick fir trees—when the margin of matching error was 2 m. In the present study, only healthy fir trees were considered, and Algorithm 1 achieved 90.81% matching. The results of Algorithm 1 showed its ability to find a high number (81.8%) of close (1 m) matches even for the NTAF dataset, proving that a dedicated algorithm can make up for some of the imprecision in the data. This higher matching quality was particularly clear for the TAF data, and was further illustrated by the percentage of predicted points that were matched to more than one real point. The percentage of repeated points increased with the margin of error ($\varepsilon$), because having further matches also means that predicted treetops can be close enough to more than one ground-truth treetop. However, in the case of Algorithm 1 and the TAF, there was a small increase that remained around 3% even for $\varepsilon = 2$ m, while Algorithm 1 using the

NTAF dataset reached 6.59% repetition. The percentage of repetition for Algorithm 2 was over 6% for both datasets (TAF and NTAF).

Algorithm 1 was much more sensitive to height variations in the CHM and, thus, missed fewer points. This came at the cost of sometimes detecting false treetops from irregularities in the canopies of fir trees or spurious elevations in the lower parts of the crowns produced by nearby deciduous trees. Our results showed how this algorithm predicted 11% of extra treetops. Algorithm 2 operated by smoothing out the boundaries of the tree crowns, but either appeared to favor single trees or failed to separate groups of trees Thus, Algorithm 2 tended to underestimate the number of treetops present, and missed some of the existing ones. This is expressed by the positive values in the "point diff" criterion (Table 2).

## 5. Conclusions

In this work, we studied the effect that the TAF has on the quality of UAV-acquired data. Our data were produced using an inexpensive UAV and publicly available elevation data. We provided qualitative and quantitative evaluation of two algorithms using TAF and NTAF datasets to automatically detect treetops. The results show that even in mountainous terrain conditions such as those presented in this study, most of the existing treetops were detected.

The results showed that Algorithm 1 was able to detect 86.55% of treetops for the TAF and 81.80% for the NTAF when only a 1 m margin of error was set. Thus, fewer points matched the ground-truth treetop more than once when the TAF was used. Treetop detection was improved by 14% when using Algorithm 2 and the TAF (76.23%) compared to when using the NTAF (62.06%) for a 1 m margin of error. Thus, our study showed that using the TAF on the acquisition of UAV data decreased matching repetition and improved treetop detection by providing better CHMs.

**Author Contributions:** O.B.H.G., L.H.S., Y.D., H.T.N. and M.L.L.C. conceived the conceptualization and methodology, and supported the writing, review, and editing; O.B.H.G., L.H.S. and Y.D. developed the software, and performed the validation and investigation; O.B.H.G., L.H.S., Y.D. and M.L.L.C. wrote the original draft; O.B.H.G. and Y.D. carried out formal analysis; O.B.H.G. and M.L.L.C. administrated the data; M.L.L.C. and Y.D. provided resources and administrated the project. All authors have read and agreed to the published version of the manuscript.

**Funding:** This research received no external funding.

**Informed Consent Statement:** Not applicable.

**Data Availability Statement:** Not applicable.

**Acknowledgments:** The authors of this paper wish to thank Hiroaki Funatsu from the Tohoku Regional Forest University for his support in facilitating our fieldwork in the Zao Mountains.

**Conflicts of Interest:** The authors declare no conflict of interest.

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
