# Peer review of "Treetop Detection in Mountainous Forests Using UAV Terrain Awareness Function"

_computation, doi:10.3390/computation10060090_

Round 1
Reviewer 1 Report
The work deserves to be published in Computation magazine. the reviewer suggests to the authors to increase the bibliography of the introductory part with an increase of publications reporting the use of UAS-UAV systems in the forest.
To improve the reference bibliography in the use of CHM models not only in the forest but also in tree crops such as vines and olives.
Some corrections
line 73-DPC: acronym explained in the summary but not in the introduction
line 369 - rewrite the title of the work in lowercase
Author Response
The work deserves to be published in Computation magazine. the reviewer suggests to the authors to increase the bibliography of the introductory part with an increase of publications reporting the use of UAS-UAV systems in the forest.
To improve the reference bibliography in the use of CHM models not only in the forest but also in tree crops such as vines and olives.
Thank you for your suggestion
Some corrections
line 73-DPC: acronym explained in the summary but not in the introduction
We have added it in the reviewed manuscript (line 75).
Reviewer 2 Report
This is a study that may find interest from those involved in forest surveys from small UAS. The following are notes including issues that should be addressed and general comments.
This is about half as long as an average paper with the same scope. Many details should be added to make the paper a more valued contribution (some examples noted below). Paper should be formatted to meet the journal standards. The English should be closely scrutinized for spelling and grammar (some examples below).
Comparison is only moderately compelling, particularly when using results from the superior algorithm (1), difference of 1-5%. A more dramatic case study, or additional case studies with more terrain undulation would likely better separate TAF from NTAF.
- Line 29: The term “general purpose” implies that Algorithm 2 captures more information beyond treetop detection in contrast with Algorithm 1. Suggest stating it is a simple CV algorithm.
- Line 32: Suggest being explicit and naming the additional parameters collected that require TAF.
- Line 43: should be “tools”
- Line 81: Should be “produced”
- Line 81: DPC should be redefined outside of the abstract.
- Line 82 and throughout: Once the initialism has been introduced, it should be consistently used (TAF).
- Line 82: “algorithms”
- Line 99: “numerical” should be “mechanical”
- Line 101: Should be “GPS”
- Line 110: 281 to 285 images is not a big difference. How would flying lower with NTAF affect the results? What were the min and max heights (or distance of cameras to sparse point cloud) of the UAS for NTAF (and TAF)? Histograms of the heights for both would help.
- Line 131: How many trees were manually annotated?
- Line 140: Should be : ”object”
- Line 141 and Figure 4: What were the average GSDs for each region parts?
- Figure 4: Is this used anywhere outside of point density comparison? Figure 7 and related text discusses "lower" area. Is this the same as the "bottom" area?
- Line 170: What value was the areal threshold?
- Line 171: What is the criterion for being “large enough”?
- Line 218: What is meant by “absolute value was used to avoid errors cancelling each other out”? There are different consequences for overestimating and underestimating counts. It would be valuable to know in which direction the algorithm skews. It seems this is done per Table 2. Regardless, why not use true positive, false positive, false negative, recall, and F-scores for a more rigorous comparison? E.g.
- Mohan, M.; Silva, C.; Klauberg, C.; Jat, P.; Catts, G.; Cardil, A.; Hudak, A.; Dia, M. Individual Tree Detection from Unmanned Aerial Vehicle (UAV) Derived Canopy Height Model in an Open Canopy Mixed Conifer Forest. Forests 2017, 8, 340
- Goutte, C.; Gaussier, E. A probabilistic interpretation of precision, recall and F-score, with implication for evaluation. In Proceedings of the European Conference on Information Retrieval, Compostela, Spain, 21–23 March 2005; Springer: Berlin/Heidelberg, Germany, 2005; pp. 345–359.
- Sokolova, M.; Japkowicz, N.; Szpakowicz, S. Beyond accuracy, F-score and ROC: A family of discriminant measures for performance evaluation. In Proceedings of the Australasian Joint Conference on Artificial Intelligence, Auckland, New Zealand, 1–5 December 2008; Springer: Berlin/Heidelberg, Germany, 2008; pp. 1015–1021.
- Table 1. Why is the label for “Matching%” where it is? Should it be two rows higher?
Author Response
Thank you for your comment and suggestions.
This is a study that may find interest from those involved in forest surveys from small UAS. The following are notes including issues that should be addressed and general comments.
This is about half as long as an average paper with the same scope. Many details should be added to make the paper a more valued contribution (some examples noted below). Paper should be formatted to meet the journal standards. The English should be closely scrutinized for spelling and grammar (some examples below).
- We have taken care on the format of the manuscript to meet the standards of the journal. We have also added some new references in the introduction and added some of the contribution mentioned by the reviewer, which in our opinion has improved the explanation of our results.
Comparison is only moderately compelling, particularly when using results from the superior algorithm (1), difference of 1-5%. A more dramatic case study, or additional case studies with more terrain undulation would likely better separate TAF from NTAF.
- We agree with the assessment of the reviewer. We were also somehow surprised of the small difference by using TAF and NTAF but looking at the results we can also conclude that depending on the slope characteristics, it might not be necessary to apply TAF flight plans if the objective is just treetop detection. In our next study, we will aim at much steeper slopes in order to find the threshold to which TAF is necessary in order to detect treetops with a reasonable degree of accuracy.
Line 29: The term “general purpose” implies that Algorithm 2 captures more information beyond treetop detection in contrast with Algorithm 1. Suggest stating it is a simple CV algorithm.
- Thank you for the suggestion, we have changed it.
Line 32: Suggest being explicit and naming the additional parameters collected that require TAF.
- We have added the parameters (line 32-33).
Line 43: should be “tools”
- Corrected (line 44).
Line 81: Should be “produced”
- Corrected (line 84).
Line 81: DPC should be redefined outside of the abstract.
- We have redefined it in line 75.
Line 82 and throughout: Once the initialism has been introduced, it should be consistently used (TAF).
- We have taken care to be consistent in the reviewed manuscripts.
Line 82: “algorithms”
- This has been corrected (line 85)
Line 99: “numerical” should be “mechanical”
Thank you for the comment. It helps to check the technical specification of the drone use.
In contrast to some drones (Phantom 4 RTK) that have both numerical and mechanical shutter, DJI Mavic 2 Pro uses only a numerical shutter system.
Line 101: Should be “GPS”
- This has been corrected (line 105).
Line 110: 281 to 285 images is not a big difference. How would flying lower with NTAF affect the results? What were the min and max heights (or distance of cameras to sparse point cloud) of the UAS for NTAF (and TAF)? Histograms of the heights for both would help.
- The mission was set to take images at regular time intervals. Even do TAF applied the terrain variation parameter, the waypoints and the shutting frequency stayed unchanged (TAF or NTAF). However, on TAF, the drone spends some time going up and down that explains a slight increase in the images number.
The height above ground at take-off point varied from 79.5 to 100.5m for TAF and from 85 to 157m for NTAF. The range was 21 and 72 m for TAF and NTAF respectively. We have added that information in the manuscript (line 114-1015).
We flew at 90m lower than most of the articles (120m). flying lower will decrease the Ground Sampling Distance but will create a less sharp CHM challenging for treetop detection.
Line 131: How many trees were manually annotated?
- We had manually annotated 464 fir trees (line 219).
Line 140: Should be:”object”
- This was corrected (line 145).
Line 141 and Figure 4: What were the average GSDs for each region parts?
- TAF average GSD for the upper, middle and bottom regions was 1.99, 1.96 and 2.06cm/px with a range of 0,1. in contrast NTAF in the same order was 2.49, 2.91 and 3.35cm/px for a range of 0.86. We have added this it in the manuscript (line 147-149).
Figure 4: Is this used anywhere outside of point density comparison? Figure 7 and related text discusses "lower" area. Is this the same as the "bottom" area?
- The idea was treetop detection result validation but finally we used it only for point density comparison. Lower and bottom are the same, we have corrected it for only Bottom.
Line 170: What value was the areal threshold?
- The value was epsilon/5 (line xx).
Line 171: What is the criterion for being “large enough”?
- This was a mistake. We have clarified in the reviewed version (line 180
Line 218: What is meant by “absolute value was used to avoid errors cancelling each other out”? There are different consequences for overestimating and underestimating counts. It would be valuable to know in which direction the algorithm skews. It seems this is done per Table 2.
- What we meant was that when we used the same parameter configuration to evaluate different sites (results not included in the paper) we took the errors in absolute value to avoid missing 10 trees in one site and overpredicting 10 trees in another add up to a "perfect" prediction. However, since this is not relevant to the current manuscript, we have deleted the sentence.
Regardless, why not use true positive, false positive, false negative, recall, and F-scores for a more rigorous comparison? E.g.
- Regarding the use of the TP, FP, FN, r, p and F-score metrics, we considered that in this case they were not totally descriptive. Each of our predicted points can either be placed close enough or too far (depending on the defined epsilon) to the ground truth (GT) points to be considered matched. However, this does not deal with the situations when a single predicted point is close enough (1 m, 1.5 m or 2 m) to more than one GT point or when more than one predicted point is close to the same GT point. We considered extending the TP and FP definitions to apply it for our case but we felt that it was not possible to extend this definition in a way that appropriately reflected the importance of tree counting in the context of this research. Consequently, we designed our evaluation metrics based on these issues (number of matched points, counting measure and percentage of points with repeated matches). In order to highlight this, we have added the following sentence to the paper. "Even though it would be possible to define a matched predicted point as a true positive prediction and an unmatched one as a false positive and use these labels to use well-established metrics such as sensitivity, specificity, F-score [20], this definition would not take into account multiple matchings from predicted points to ground truth points or vice-versa. Taking into account that tree counting is an important problem in our application scenarios, we decided to use the aforementioned measures that target treetop counting. (Line 232 – 236). Citation [21] is Mohan et al., 2017, in the part of the manuscript.
Table 1. Why is the label for “Matching%” where it is? Should it be two
- We have corrected this.

Reviewer 3 Report
This study makes a comparison analysis between following the terrain and not following the terrain flight for the treetop detection. This manuscript seems to have gone through a review round already. And careful revisions can be found in this article. However, some problems have still existed. Therefore, I recommend the manuscript should make another revision at this stage.
Major comments:
- In general, this work is more like a 'technical report' instead of a 'scientific paper'. There are too many technical details. I suppose the authors could discuss more about the algorithm details instead of how to use the software.
- In the Introduction section, the authors need to define the importance and the novelty of their work more clearly. Specifically, the first paragraph is too lengthy. I suggest the authors split the first paragraph into two parts.
- A statement appears in Line 82, “...which in turn can contribute significantly to Deep Learning application on forestry”, which makes me confused. Since I didn’t find out this work used DL method to detect treetop. This issue should be clarified.
- Another confusing point appears in Figure 1. Numbers in the scale bar in the lower left corner should be confirmed. I think there may be something mistakes in the spatial reference. Please check.
- In the Methodology section, I suggest the authors describe two algorithms more clearly, especially algorithm 1. A flowchart of this algorithm should be considered to make the related contents more clear. Besides, a necessary equation could be demonstrated. Some important metrics, such as the value of epsilon/5, need more explanations.
Minor comments:
- Figure 6 from (a) to (e) all look the same. Please make these pictures more clear.
- Please use the official template for this article.
Author Response
Major comments:
In general, this work is more like a 'technical report' instead of a 'scientific paper'. There are too many technical details. I suppose the authors could discuss more about the algorithm details instead of how to use the software.
We are really sorry that our submitted manuscript gave that impression. In the modified version we emphasize the explanation and performance of the algorithms. We have also erased explanation that was indeed too technical.
In the Introduction section, the authors need to define the importance and the novelty of their work more clearly. Specifically, the first paragraph is too lengthy. I suggest the authors split the first paragraph into two parts.
A: We have followed the advice of the reviewer and split the first paragraph of the introduction and also erased some text to make the explanation clearer.
The novelty of this study is that we have tested the effect of TAF on UAV forestry application, in particular on treetop detection, which is at present one of the most important issues in forest research. There are few studies who have mentioned this issue [18, 19] but they have only evaluated NTAF datasets. In fact, there is no mention of the slope effect in most of the studies aimed at predicting treetop detection in forests.
A statement appears in Line 82, “...which in turn can contribute significantly to Deep Learning application on forestry”, which makes me confused. Since I didn’t find out this work used DL method to detect treetop. This issue should be clarified.
A: We erased this sentence because it does not contribute to the present paper.
Another confusing point appears in Figure 1. Numbers in the scale bar in the lower left corner should be confirmed. I think there may be something mistakes in the spatial reference. Please check.
A: This was an error. We have corrected the spatial reference.
In the Methodology section, I suggest the authors describe two algorithms more clearly, especially algorithm 1. A flowchart of this algorithm should be considered to make the related contents clearer. Besides, a necessary equation could be demonstrated. Some important metrics, such as the value of epsilon/5, need more explanations.
A: We have followed the advice of the reviewer and have added more explanation about the algorithms used in this study. Line 229-232
Minor comments:
Figure 6 from (a) to (e) all look the same. Please make these pictures clearer.
A: We have modified this figure so the meaning of it can be clearly understood.
Please use the official template for this article.
A: We have made sure that our manuscript adjusts to the template of the Journal.

Round 2
Reviewer 2 Report
I appreciate the responses that the authors provided. However, in my opinion this paper still needs a more compelling scenario, steeper terrains, to warrant publication. Even better would be inclusion of different scenes. The responses indicate the authors intend on further study to do this. I would suggest including the data from this manuscript combined with that from the next study. This will make it a better paper covering a more acceptable scope for a refereed journal article.
Author Response
I appreciate the responses that the authors provided. However, in my opinion this paper still needs a more compelling scenario, steeper terrains, to warrant publication. Even better would be inclusion of different scenes. The responses indicate the authors intend on further study to do this. I would suggest including the data from this manuscript combined with that from the next study. This will make it a better paper covering a more acceptable scope for a refereed journal article.
We agree with the reviewer that the slope we used in our experiment was not steep enough to produce more significant treetop detection differences between the application of two algorithms on TAF and NTAF datasets. However, there was an improvement, especially of 14% when Algorithm 2 was used. In our study, the slope we used had 20 degrees steepness and it cannot be considered as a mild slope. As mentioned above a sharper slope will probably show a clearer difference when using TAF and NTAF but as we point out in our study the algorithm also plays an important role.
The novelty of this study is that we have tested the effect of TAF on UAV forestry application, in particular on treetop detection, which is at present one of the most important issues in forest research. There are few studies who have mentioned this issue [18, 19] but they have only evaluated NTAF datasets. In fact, there is no mention of the slope effect in most of the studies aimed at predicting treetop detection in forests.
Furthermore, as we have shown in the manuscript, even though our objective is treetop detection, the results showed that the image quality of individual tree crown characteristics improved with TAF. These results can be used for the analysis of individual trees conditions such as health, damage, growth and many other physiological aspects of trees that are important to understand the temporal and spatial effect of climate change on forest ecosystems.
Thus, we considered that the results of this study represent an initial step for more studies in the improvement of the quality of datasets obtained with UAVs in mountainous areas. As mentioned by the reviewer we could add more and steeper slopes but they will have to be with the same tree species since different tree species and the characteristics of that particular slope will add more complexity to the comparison that has to be properly addressed.
However, in our present study, particularly fir forests are found in the range of slopes or milder than the one presented in this study and thus, the results of this study highly contribute to further survey of this area. It appears that the steeper the slope, the larger can be the distortion of the CHM during DPC normalization, since the treetop shifts position towards the upper side of the slope.
The present study is part of a much larger study focused on fir tree health status detection in mountainous areas of Yamagata Prefecture in Japan.
Reviewer 3 Report
Thanks to the authors provided the revised manuscripts. It seems like all of my questions have been answered. Here are some minor issues that can be improved.
1. I think Figure 8 should be better changed from a screenshot figure to a table.
2. Figure 9 should add a scale bar.
Author Response
Thank you very much for you thorough review.
We have apply your recommendation and change the Figures.